DOI: 10.1038/s41467-017-00245-9　　OPEN

# Gap-state engineering of visible-light-active ferroelectrics for photovoltaic applications

Hiroki Matsuo[1,3], Yuji Noguchi [2] & Masaru Miyayama [2]

Photoferroelectrics offer unique opportunities to explore light energy conversion based on their polarization-driven carrier separation and above-bandgap voltages. The problem associated with the wide bandgap of ferroelectric oxides, i.e., the vanishingly small photo-response under visible light, has been overcome partly by bandgap tuning, but the narrowing of the bandgap is, in principle, accompanied by a substantial loss of ferroelectric polarization. In this article, we report an approach, 'gap-state' engineering, to produce photoferroelectrics, in which defect states within the bandgap act as a scaffold for photogeneration. Our first-principles calculations and single-domain thin-film experiments of $BiFeO_3$ demonstrate that gap states half-filled with electrons can enhance not only photocurrents but also photo-voltages over a broad photon-energy range that is different from intermediate bands in present semiconductor-based solar cells. Our approach opens a promising route to the material design of visible-light-active ferroelectrics without sacrificing spontaneous polarization.

[1] Department of Advanced Interdisciplinary Studies, School of Engineering, The University of Tokyo, Tokyo 113-8656, Japan. [2] Department of Applied Chemistry, School of Engineering, The University of Tokyo, Tokyo 113-8656, Japan. [3]Present address: Department of Chemical System Engineering, School of Engineering, The University of Tokyo, Tokyo 113-8656, Japan. Correspondence and requests for materials should be addressed to Y.N. (email: ynoguchi@fmat.t.u-tokyo.ac.jp)

Innovative strategies have recently been introduced into photovoltaics for efficient solar energy conversion[1–4]. To absorb light over a broad spectral range and convert the incident energy into electricity, intermediate band (IB) solar cells are designed in which the IB provides electronic states that are partially filled with electrons within the bandgap of a host semiconductor. The IB acts as a stepping stone to generate an electron–hole pair under illumination with below bandgap energy, thereby delivering a large photocurrent[5–7]. In spite of extensive studies on IB solar cells, the present devices suffer from a reduction in open-circuit voltage compared with the reference cells without IBs[8].

The photovoltaic (PV) effect in polar materials[9–11] has allowed researchers to develop novel methods of light energy conversion, generating voltages beyond the bandgap limit of semiconductor p–n junctions[12, 13]. Because ferroelectrics have multiple polar states[14] that can be selectively stabilized by electric and/or stress fields and exhibit a photoresponse arising from spatial inversion symmetry breaking[15, 16], advanced functionalities based on the coupling between the PV effect and the switchable polarization[17, 18] have stimulated renewed interest in photoferroelectrics[19, 20].

The ferroelectric PV effect has been extensively studied for LiNbO$_3$[9, 21, 22], BaTiO$_3$[23–27], PbTiO$_3$-based perovskites[28–30] and BiFeO$_3$ (BFO)[12, 13, 31, 32]. Recent studies have demonstrated that control of domain architecture can enhance the photoresponse due to the domain-wall-driven effect[12, 27]. However, these conventional ferroelectrics suffer from poor absorption of visible light because of their bandgap energy ($E_g$) of ~3–4 eV[33–35]. Even for BFO with its relatively small $E_g$ of 2.7 eV[36, 37], photons with wavelengths of $\lambda > 460$ nm, constituting >80 % of the sunlight, cannot be utilized, raising the question of how to activate the PV effect under visible light.

In ferroelectric oxides, their electronic structure with a wide bandgap is characterized by an O-2$p$-derived valence band (VB) and a transition metal (TM)-$d$-derived conduction band (CB) that are accompanied by the orbital interactions of cations and oxygen atoms[38, 39]. The TM-$d$ and O-2$p$ hybridization is enhanced upon atomic displacements[40], and the hybridization of Pb- or Bi-6$s$($p$) with O-2$p$ induces strong covalent bonding associated with polar lattice distortions[14, 41]. Both of these cation–oxygen interactions favour coherent off-centre displacements of the cations in their oxygen polyhedra, delivering substantially large spontaneous polarization ($P_s$)[42].

Considerable effort has been devoted to narrowing the bandgap of ferroelectrics by rational modification of their compositions. Bandgap tuning has produced narrow-bandgap materials such as KBiFe$_2$O$_5$[43] with FeO$_4$ tetrahedra, site-specific LaCoO$_3$-substituted Bi$_4$Ti$_3$O$_{12}$[44], heavily Co- and Fe-doped Bi$_{3.25}$La$_{0.75}$Ti$_3$O$_{12}$[45], KNbO$_3$-based solid solution of [KNbO$_3$]$_{1-x}$ [BaNi$_{0.5}$Nb$_{0.5}$O$_{3-\delta}$]$_x$[46], and cation-ordered Bi(Fe,Cr)O$_3$[47], exhibiting enhanced PV responses under visible light. This bandgap tuning is based on a solid solution or structural formation composed of wide-gap ferroelectrics and non-polar counterparts and is, in principle, accompanied by a substantial reduction in $P_s$. Meanwhile, the strongly distorted polar lattice yields a robust photoresponse[15, 16], called the bulk PV (BPV) effect[9–11], and the high photovoltages relies on the large magnitude of $P_s$. The tradeoff between narrow gap and large polarization motivates us to explore an alternative route to the material design of photoferroelectrics that exhibit a response to low-energy light without sacrificing $P_s$.

Here, we report an approach, 'gap-state' engineering, to produce visible-light-active ferroelectrics. Our approach is based on the creation of defect states within the bandgap, i.e., gap states, half of which are filled with electrons. Under illumination with below-bandgap energy, photon absorption via the half-filled states enables the empty states to receive electrons from the VB and the filled states to supply them to the CB, generating electron–hole pairs. We choose BFO as a model material to investigate the impact on the BPV effect. Our theoretical and experimental investigations demonstrate that the gap states enhance both photocurrents and photovoltages over a broad

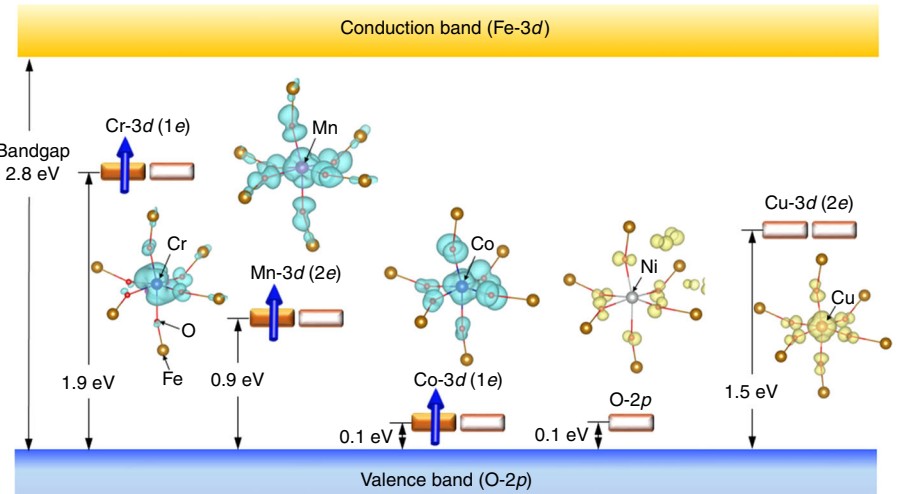

**Fig. 1** Wave functions and schematic electronic structures of transition metal (TM=Cr, Mn, Co, Ni and Cu)-doped BFO obtained from DFT calculations. The bandgap ($E_g$) of BiFeO$_3$ (BFO) is calculated to be 2.8 eV. The valence band maximum (VBM) is primarily formed by O-2$p$, and the conduction band minimum (CBM) is formed by Fe-3$d$. Filled and open rectangles indicate electron-filled and electron-unoccupied states, respectively. The $e$ states of Cr, Mn and Co are half-filled with electrons, while the $e$ state of Cu is empty. The blue arrows designate spin-polarized electrons in the gap states. Except for TM=Ni, the gap states are derived from the 3$d$ orbitals of TMs hybridized with the surrounding O-2$p$. For TM=Ni, the 3$d$-derived filled states are present below the VBM, and the unoccupied O-2$p$ state appears just above the VBM, showing Ni$^{2+}$ with a ligand hole in the O-2$p$ state (see text). The wave functions of each gap state are visualized. The wave functions coloured in blue feature orbitals filled with electrons, and those in yellow present empty orbitals, where the iso-surface level remains constant for displaying the wave functions. The 3$d$ orbitals of TMs are hybridized with the O-2$p$ of the surroundings in different manners reflecting their interactions. The 1$e$ state of TM=Cr and Co shows $\pi^*$ bonding having $d_{xy}$ and $d_{x2-y2}$ characters, while the 2$e$ state of TM=Mn and Cu exhibits $\sigma^*$ bonding featuring $d_{yz}$ and $d_{zx}$ characters

spectral range. This approach opens a promising route to the development of visible-light-active ferroelectrics with little or no loss of $P_s$.

## Results

**Electronic structure calculations**. Figure 1 shows the schematic band structures of the TM-doped BFO. The $E_g$ of BFO is calculated to be 2.8 eV, consistent with experiments (2.7 eV[36, 37]). The TM doping does not change significantly in either the $E_g$ or in the electronic character of the VB maximum (VBM) constructed by the O-$2p$ orbitals and the CB minimum (CBM) formed by the Fe-$3d$ orbitals (Supplementary Fig. 1).

According to group theory, the $3d$ orbitals of the TMs in rhombohedral BFO ($C_{3v}$ symmetry) are split into $a_1$, $1e$ and $2e$, where $1e$ and $2e$ denote lower-lying and higher-lying $e$ states, respectively. As shown in Fig. 1, the TMs of Cr, Mn, Co and Cu produce $e$ states within the bandgap. Especially, Cr, Mn and Co yield the gap states, half of which are filled with electrons (half-filled). Relative to the VBM, the gap state is located at 1.9 eV for Cr and 0.9 eV for Mn, while that of Co is in the vicinity of the VBM.

Figure 2 displays the detailed electronic structure of Mn (6%)-doped BFO with the optimized structure, the density of states and the band structure. A partial density of states analysis reveals that

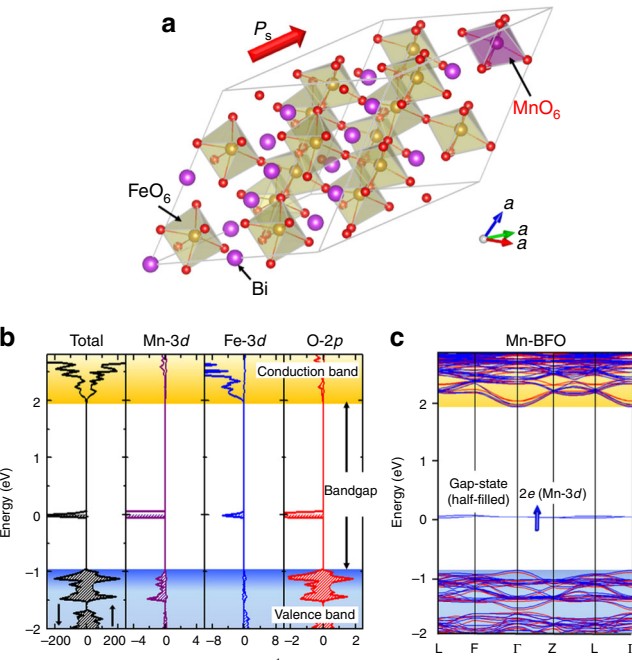

**Fig. 2** Crystal and electronic structures of Mn-doped BFO. **a** Optimized crystal structure of Mn-doped BFO [Bi$_{16}$(Fe$_{15}$Mn)O$_{48}$] in rhombohedral $R3$ symmetry, **b** total density of states (DOS) and partial DOS (PDOS) of Mn-$3d$, Fe-$3d$ and O-$2p$ and **c** electronic band structure. In **b**, the nearest neighbour Fe and O atoms of Mn are selected, and majority- and minority-spin bands are described in the *right* ($\uparrow$) and *left* ($\downarrow$) panels, respectively. The Mn-$3d$ orbitals are split into up-spin $a_1$, 1e and 2e and down-spin $a_1$, 1e and 2e. Taking the $d^4$ electron configuration of Mn$^{3+}$ into account, we can regard it as up-spin $a_1^1$, $1e^2$ and $2e^1$ and down-spin $a_1^0$, $1e^0$ and $2e^0$. In **c**, red and blue lines represent majority- and minority-spin states, respectively, and the blue arrow indicates a spin-polarized electron in the gap state, exhibiting that Mn provides the half-filled gap (2e) state. For the 2e state, the $d_{yz}$- and $d_{zx}$-derived orbitals are degenerate at the $\Gamma$ and $Z$ points in the Brillouin zone, while the degeneracy is lifted once the $k$-point is away from these high-symmetry points; its energy difference is 0.03 eV at the L point

the gap state arises from the hybridized orbital of the Mn-$2e$ and the adjacent O-$2p$ with a significant contribution of Fe-$3d$, as shown in its wave function (Fig. 1). The $2e$ state is composed of the $d_{yz}$- and $d_{zx}$-derived orbitals (Fig. 2c) and their spin-polarized electron occupancies are in the range of 0.4–0.6 over the entire Brillouin zone. We can therefore regard the $2e$ as a half-filled state (see Supplementary Note 2).

We found that Ni on the Fe site has a $d^8$ electron configuration, representing the Ni$^{2+}$ valence state. The $d$-derived state of Ni$^{2+}$ does not construct a gap state, while an unoccupied O-$2p$ state appears just above the VBM. This result indicates that one hole is transferred from Ni to the ligand oxygen in the NiO$_6$ octahedron, as reported in ref. [48]. This electronic state is expressed as Ni$^{2+}L(d^8L)$, where $L$ denotes a ligand hole.

Under illumination with $h\nu$ (photon energy) $>E_g$, photons pump electrons from the VB to the CB and generate electron–hole pairs that are spatially separated in the polar lattice. The optical transition associated with this charge separation delivers a PV current. Given that electronic states within the bandgap, so-called 'gap' states, are present, the absorption of light at $h\nu<E_g$ is capable of inducing a PV response. For gap states filled with electrons, pumping electrons from the gap states to the CB injects carriers leaving behind empty traps, and the continuous illumination yields a PV current arising from a sequential trapping and detrapping of photo-generated electrons[21, 22, 49]. When gap states are empty, pumping electrons from the VB to the gap states creates holes, and this illumination also brings a PV current via trapping and detrapping of photogenerated holes[26].

Provided that the gap states are partially filled with electrons, we expect that the absorption of two sub-bandgap photons generates electron–hole pairs by pumping electrons from the VB to the CB via the gap states, as is seen for IB solar cells[3, 8]. This two-photon absorption has a high probability when its state is half-filled[50]. Moreover, its energy level is preferred in or near the middle of the bandgap. Compared with the filled or empty gap states, where the majority photocarrier is either electron or hole, respectively, we anticipate that the half-filled mid-gap state acts as a scaffold for a robust PV response under illumination with below-bandgap light.

The material design strategy described above leads us to hypothesize that the TMs of Cr and Mn are promising candidates for enhancing the PV effect. Because of a small solubility limit of Cr (several percentage[51]), we think that Mn is attractive as the TM dopant for two reasons: there is practically no solubility limit, and its doping does not deteriorate $P_s$ up to 50%[52]. Here, we choose Mn as a dopant for testing the hypothesis of the half-filled gap state and, for comparison, Ni as an example of how the empty state plays a role in the BPV effect.

**Photocurrent properties**. In Fig. 3a, we illustrate the single-domain (SD) structure associated with the pseudocubic [$hkl$] directions showing the crystallographic orientations (also see Supplementary Fig. 2). For a precise description of the 4° vicinal angle from the pseudocubic axes owing to the miscut of the substrate, we employ the lab-coordinate ($xyz$) system exhibiting the PV measurement configuration (Fig. 3b). Figure 3c presents the $J$–$V_{bias}$ properties in the $y$ direction under illumination ($h\nu = 2.4$ eV) at $\omega = 90°$ ($\omega$ denotes light polarization angle). The Mn-BFO film shows a $V_y$ of $-3.1$ V and a $J_y$ of $-15$ μA cm$^{-2}$ that are much higher than those of the BFO film ($V_y = -0.6$ V, $J_y = -1.3$ μA cm$^{-2}$).

In Fig. 3d, e, we exhibit the $J_y$ and $J_{-x}$ data, respectively, as a function of $\omega$ at $h\nu = 2.4$ eV for the Mn-BFO film. We found that $J_{-x}$ (Fig. 3e) shows a sinusoidal dependence on $\omega$ without offset,

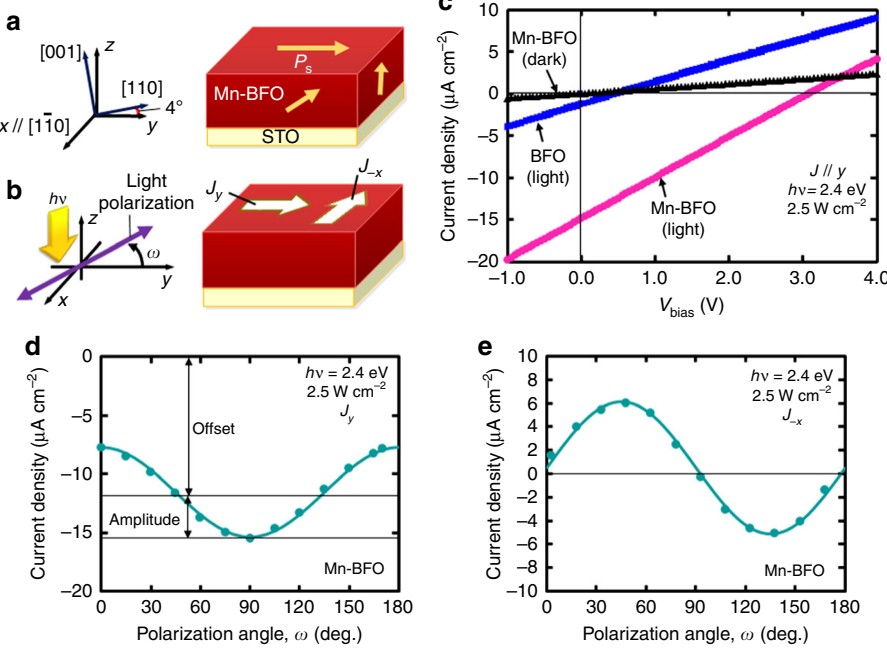

**Fig. 3** Measurement configuration and photovoltaic (PV) properties of the Mn-BFO film. **a** Schematic of the single-domain (SD) structure associated with the pseudocubic [$hkl$] directions exhibiting the crystallographic orientations with respect to the lab-coordinate ($xyz$) system. The $x$ axis lies along [$1\bar{1}0$], and the $y$ and $z$ axes are 4° away from [110] and [001], respectively. **b** PV measurement configuration in which polarized light with photon energy $h\nu$ propagates along the $z$ axis and is incident on the sample surface. The incident light has a polarization angle ($\omega$) from the $y$ axis in the $xy$ plane. The short-circuit current density ($J_{sc}$) and open-circuit voltage ($V_{oc}$) along the $m$ direction are expressed as $J_m$ and $V_m$, respectively. A positive $J_m$ indicates that the photocurrent flows in the $m$ direction. The bias voltage ($V_{bias}$) at which the current density ($J$) becomes zero is denoted by $-V_m$. **c** Current density–bias voltage ($J$–$V_{bias}$) properties along the $y$ direction under dark and illuminated ($h\nu$=2.4 eV) conditions, where the polarization angle $\omega$ is aligned so that the PV response becomes maximum for both the BFO and Mn-BFO (Mn 5%) films. Light polarization angle $\omega$ dependences of short-circuit currents **d** $J_y$ and **e** $J_{-x}$ in the Mn-BFO film

while $J_y$ (Fig. 3d) has an offset of $J_{y,off.}$ in addition to a sinusoidal oscillation with an amplitude $J_{y,amp.}$ (the data of the BFO film are presented in Supplementary Fig. 4c, d). Even at $h\nu$=1.9 eV, the Mn-BFO film clearly shows similar sinusoidal behaviour in $J_y$ and $J_{-x}$ (Supplementary Fig. 5c, d, whereas we cannot observe a PV response for the BFO film.

**Photon energy dependence.** In Fig. 4a, b, we show the $h\nu$ dependences of $|V_y|$ and $|J_y|$, respectively. Under illumination at $h\nu$=3.1 eV, the BFO film generates a $|V_y|$ of 4.3 V. However, $|V_y|$ decreases to 0.6 V at $h\nu$=2.4 eV and below the detection limit at $h\nu$=1.9 eV (Fig. 4a). A similar reduction in $|J_y|$ is also detected in the low-$h\nu$ range (Fig. 4b). We note that both the $|V_y|$ and $|J_y|$ of the Mn-BFO film are much larger than those of the BFO film at $h\nu$ above $E_g$ in addition to below $E_g$. The Mn-BFO film exhibits a $|V_y|$ exceeding the bandgap voltage under illumination at $h\nu$=2.4 eV. The Mn-BFO film clearly exhibits a substantial PV response at 1.9 eV and no detectable signal at 1.5 eV (Supplementary Fig. 11).The overall tendency of the PV response with respect to $h\nu$ agrees qualitatively with the results of the absorption spectra (Supplementary Fig. 3). (An estimation procedure of $\alpha$ is presented in Supplementary Note 4 and Supplementary Fig. 10.)

The Ni-BFO film displays a relatively high $|J_y|$ whereas the $|V_y|$ is lower than 0.1 V at 2.4 eV (Supplementary Fig. 8). The small $|V_y|$ is attributed to its high conductivity (Supplementary Table 3), and the resultant photogenerated power $|J_y \times V_y|/4$ becomes much smaller.We note that its high conductivity and density functional theory (DFT) calculation results of Ni-doped BFO confirm $Ni^{2+}\underline{L}(d^8\underline{L})$ state in the Ni-BFO film (see Supplementary Note 3 and Supplementary Fig. 9).

## Discussion

We estimate the BPV tensor elements ($\beta_{15}$, $\beta_{22}$, $\beta_{31}$ and $\beta_{33}$) from the $\omega$ dependences of $J_y$ and $J_{-x}$ by taking into account the 4° deviation of the measurement directions from the pseudocubic axes, the details of which are described in ref. [32]. In Fig. 5a, we present the BPV tensor elements at $h\nu$=2.4 eV obtained by fitting the experimental data with the analytic equations. We found that the absolute values of the Mn-BFO film are 2–10 times as large as those of the BFO film. This result clearly demonstrates that the Mn doping dramatically enhances the BPV effect under illumination with below-bandgap light. The apparent $\omega$-dependent photocurrents of the Mn-BFO film at $h\nu$=1.9 eV (Supplementary Fig. 5c, d) enable us to obtain the BPV tensor elements (Supplementary Fig. 7).

Figure 5b exhibits the BPV tensor elements at $h\nu$=3.1 eV in a similar manner. The Mn doping decreases $|\beta_{15}|$ and $|\beta_{31}|$ but increases $|\beta_{33}|$ markedly. This result indicates that the larger values of $|V_y|$ and $|J_y|$ at $h\nu$=3.1 eV achieved by the Mn doping (Fig. 4a, b) originate from the enhanced $\beta_{33}$. The BPV tensor elements are listed in Supplementary Table 1.

Here, we discuss why the Mn doping markedly enhances the BPV effect at $h\nu<E_g$. The BFO film shows the BPV effect at $h\nu$=3.1 eV because photons with $h\nu \geq E_g$ (2.7 eV) can generate electron–hole pairs that are subsequently separated in the polar lattice. A small but significant photoresponse of the BFO film at $h\nu$=2.4 eV is attributed to a dipole-forbidden $p$-$d$ charge transfer transition[53].

The robust response of the Mn-BFO film at $h\nu<E_g$ enables us to consider that the optical transitions via the half-filled gap state enhances the BPV effect. The Mn-BFO film at $h\nu$=2.4 eV exhibits the comparable or large BPV tensor elements compared with the

BFO film at $h\nu$=3.1 eV ($h\nu$>$E_g$), supposing that the PV effect observed at $h\nu$=2.4 eV stems from electron–hole pair generation. The details of the photogenerated carriers are described in Supplementary Note 5. In principle, visible light with $h\nu$ higher than the sub-bandgap energies can create electron–hole pairs via the gap state by the following two-photon absorption. A sub-bandgap transition pumps an electron from the gap state to the CB (a charge transfer from Mn to its adjacent Fe) and does so

from the VB to the gap state (a charge transfer from O to Mn), where these processes are accompanied by the changes in the valence state of Mn. Even when the optical transitions occur successively, electron–hole pairs are generated, because once an electron or hole is pumped the carrier is directed away from Mn in the polar lattice.

The Mn-BFO film exhibits an apparent PV response at $h\nu$=1.9 eV but no detectable short-circuit photocurrent at $h\nu$=1.5 eV (Supplementary Fig. 11). These results combined with the optical transition processes (Supplementary Note 5 and Supplementary Fig. 12) suggest that the majority photocarrier at $h\nu$=1.9 eV is electron–hole pair. In contrast, we found that photogenerated holes can induce the BPV effect in the Ni-BFO film at $h\nu$=1.9 eV (Supplementary Fig. 8). The Ni doping provides the empty state in the vicinity of the VBM. This shallow state gives rise to the high conductivities under dark and illumination (Supplementary Table 3), leading to the high currents with sacrificing the voltages. The much higher photogenerated power of the Mn-BFO film compared with the Ni-BFO film shows that the half-filled gap state is effective for the BPV effect.

Next, we discuss the origin of the enhanced $\beta$ in Mn-BFO at $h\nu<E_g$. The BPV tensor element $\beta_{ij}$ can be written as $\beta_{ij}=\alpha G_{ij}$[21], where $\alpha$ denotes the absorption coefficient and $G_{ij}$ indicates the Glass coefficient determined by the shift vector representing the average distance travelled by the coherent carriers during their lifetimes[15]. Though anisotropy in $\alpha$[54] may cause variations in the photocurrents, the experimental data of the $\omega$-dependent $J_y$ and $J_{-x}$ are well fitted by the analytical equations adopting an isotropic $\alpha$, showing that the anisotropy in $\alpha$ does not have a significant influence on our PV results.

One possible origin of the enhancement in $\beta$ by the Mn doping is an increase in $\alpha$. We found, however, that the Mn doping yields only a 22% increase in $\alpha$ at $h\nu = 2.4$ eV (Supplementary Fig. 3) while the $\beta$ of the Mn-BFO film is 2–10 times as large as that of the BFO film, demonstrating that the enhanced $\beta$ cannot be explained solely by its slightly larger $\alpha$.

We now estimate $G_{ij}$ representing photogeneration strength in the BPV effect. The one-photon transition of the VB to CB has a contribution of $\beta_{ij,1}$ (=$\alpha_1 G_{ij,1}$) to the BPV effect, as observed for the BFO film. The two-photon absorption via the gap (2$e$) state delivers an additional $\beta_{ij,2}=\alpha_2 G_{ij,2}$ for the Mn-BFO film, where $\alpha_1$ and $\alpha_2$ denote their respective absorption coefficients. Because the Mn doping does not change the fundamental feature of the VB and the CB, $\beta_{ij}$ representing the net BPV response of Mn-BFO

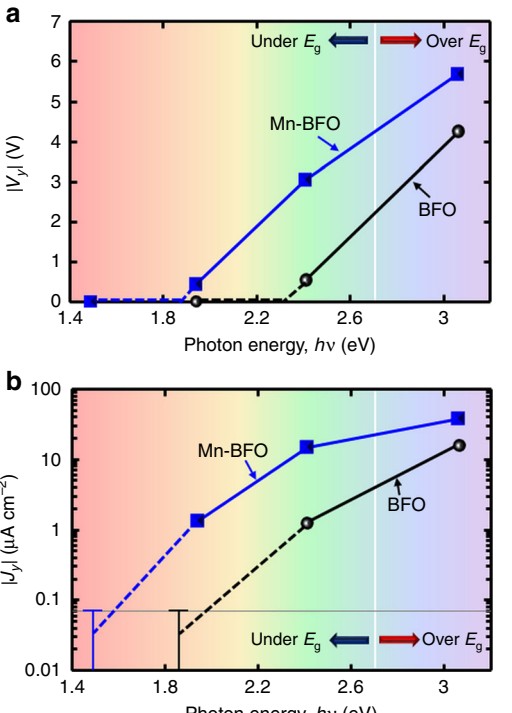

**Fig. 4** Photogenerated current and voltage as a function of photon energy ($h\nu$). **a** |$V_y$| and **b** |$J_y$|, where the polarization angle $\omega$ is aligned so that the PV response becomes maximum: $\omega = 90$ ° in the BFO film (regardless of $h\nu$); $\omega = 0$ ° at $h\nu = 3.1$ eV, and $\omega = 90$ ° at $h\nu = 2.4$ eV and 1.9 eV in the Mn-BFO (Mn 5 %) film. The horizontal line in **b** indicates the detection limit in our measurements. The $J$-$V_{bias}$ properties of the films of BFO ($h\nu$=3.1 eV) and Mn-BFO ($h\nu$=3.1 eV, 1.9 eV and 1.5 eV) in the $y$ direction are shown in Supplementary Fig. 6a, b 11, respectively

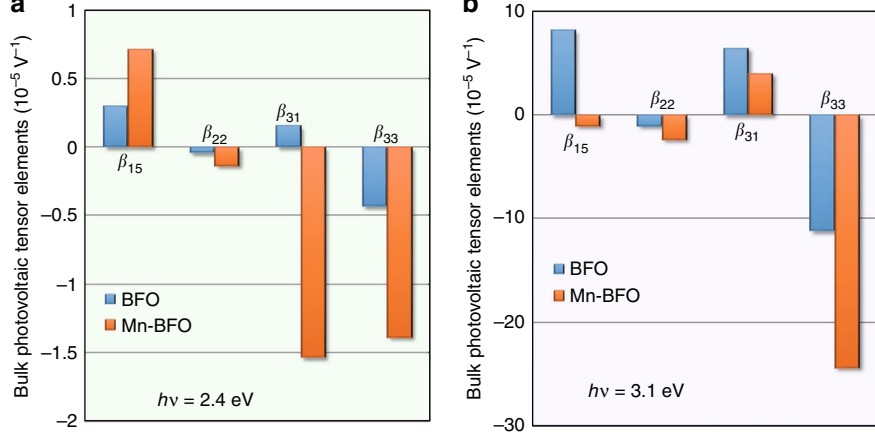

**Fig. 5** Bulk photovoltaic (BPV) tensor elements ($\beta_{ij}$). Under illumination at **a** $h\nu$=2.4 eV and **b** $h\nu$=3.1 eV estimated from light polarization angle $\omega$ dependences of short-circuit currents. The data of the BFO film are shown in Supplementary Fig. 4 and that of the Mn-BFO film are exhibited in Supplementary Fig. 5

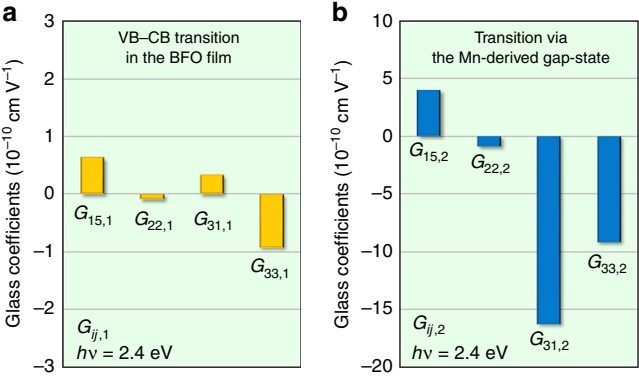

**Fig. 6** Glass coefficients of $G_{ij,1}$ and $G_{ij,2}$ at $h\nu=2.4$ eV. **a** $G_{ij,1}$ represents the photogenerated strength arising from the one-photon optical transition of the VB to the CB in the BFO film, while **b** $G_{ij,2}$ reflects what is attributed to the charge transfer transition via the Mn-3$d$-derived gap (2e) state

is expressed as $\beta_{ij}=\beta_{ij,1}+\beta_{ij,2}$. As a result, $G_{ij,2}$ arising from the Mn-derived gap state can be expressed as

$$G_{ij,2}=\frac{\beta_{ij,2}}{\alpha_2}=\frac{1}{\alpha_2}\left(\beta_{ij}^{\text{Mn-BFO}}-\beta_{ij}^{\text{BFO}}\right),\qquad(1)$$

where $\beta_{ij}^{\text{BFO}}$ and $\beta_{ij}^{\text{Mn-BFO}}$ are their respective experimental $\beta_{ij}$ values.

Using the relations of $\alpha_1=\alpha^{\text{BFO}}$ and $\alpha_1+\alpha_2\approx\alpha^{\text{Mn-BFO}}$ and the experimental result of $\alpha^{\text{Mn-BFO}}=1.22\alpha^{\text{BFO}}$ ($h\nu=2.4$ eV, see Supplementary Fig. 3), we obtain $\alpha_2=0.22\alpha^{\text{BFO}}$. From the experimentally determined $\beta_{ij}^{\text{BFO}}$ and $\beta_{ij}^{\text{Mn-BFO}}$ values (Fig. 5a), $G_{ij,1}\left(=\frac{\beta_{ij,1}}{\alpha_1}=\frac{\beta_{ij}^{\text{BFO}}}{\alpha^{\text{BFO}}}\right)$ and $G_{ij,2}$ at $h\nu=2.4$ eV are estimated as displayed in Fig. 6a, b, respectively. The $G_{ij,1}$ and $G_{ij,2}$ values at $h\nu=3.1$ eV, 2.4 eV and 1.9 eV are summarized in Supplementary Table 2, where the data at 3.1 eV and 1.9 eV are estimated in a similar manner at 2.4 eV. The above estimation reveals that $|G_{ij,2}|$ is much larger than $|G_{ij,1}|$: $|G_{31,2}|$ is ~50 times as large as $|G_{31,1}|$ at $h\nu=2.4$ eV. These results lead to the conclusion that the major origin of the enhanced PV response achieved by the Mn doping is not the increase in $\alpha$ but the large Glass coefficients.

Finally, we address how the half-filled gap states in ferroelectrics are different from IBs in semiconductor solar cells. For bulk IB solar cells, such as $V_2In_{14}S_{24}$[55] and ZnTe:O[7], extended IBs are situated in or near the middle of the bandgap. The IBs act as a stepping stone for generating electron–hole pairs by absorbing sub-bandgap photons and thereby photocurrents are enhanced by an increase in $\alpha$ at $h\nu<E_g$[8]. The crucial problem to be overcome is a reduction in $V_{oc}$ with respect to the reference cells without IBs at room temperature[8], where the Shockley–Read–Hall (SRH) recombination[56] influences their $V_{oc}$ to some extent. In contrast, the gap-state engineering in ferroelectrics can not only increase $\alpha$ but also enhance $V_{oc}$. In fact, the Mn doping leads to larger values of both $V_{oc}$ and $J_{sc}$ (Fig. 4). Although the SRH recombination is suggested to reduce the carrier lifetimes, the gap state increases the Glass coefficients by over one order of magnitude at $h\nu=2.4$ eV ($h\nu<E_g$). This experimental fact shows that the average drift velocity of photogenerated electrons and holes is higher in the presence of the gap state, and thereby delivering the robust BPV effect.

Our experimental and theoretical investigations demonstrate that the mid-gap half-filled state acts as a scaffold for the BPV effect over a broad $h\nu$ range. Gap-state engineering is revealed to enhance not only photocurrents but also photovoltages, and therefore providing a promising route to the material design of visible-light-active photoferroelectrics.

## Methods

**DFT calculations**. First-principles calculations based on DFT[57] were performed within the generalized gradient approximation (GGA)[58] in the projector-augmented-wave method[59], as implemented in the Vienna *ab initio* simulation package[60]. We employed the gradient corrected exchange–correlation functional of the PBEsol (Perdew–Burke–Ernzerhof revised for solids)[61] for structural optimization. To obtain insight into the role of the defect states arising from TM elements, accurate information on bandgaps is an essential foundation for investigating their influence on the bulk PV effect. We therefore adopted the density functional MBJLDA (modified Becke–Johnson exchange plus local density approximation)[62] to calculate the electronic band structures and density of states. Within the simplified GGA+$U$ approach[63], on-site Coulomb interaction parameters of $U-J=4$ eV were added to all the $d$ elements of the TMs, including Fe. The Monkhorst–Pack $k$-point mesh of $3\times3\times3$ was adopted for the supercell of $Bi_{16}$ ($Fe_{15}TM)O_{48}$ in rhombohedral $R3$ symmetry (TM=Cr, Mn, Co, Ni and Cu) (see Supplementary Note 1). All results were obtained by explicitly treating 15 valence electrons for Bi ($5d^{10}6s^26p^3$), 8 for Fe ($3d^64s^2$), 6 for Cr ($3d^44s^2$), 7 for Mn ($3d^54s^2$), 9 for Co ($3d^74s^2$), 10 for Ni ($3d^84s^2$), 11 for Cu ($3d^{10}4s^1$) and 6 for O ($2s^22p^4$). The plane-wave cutoff energy was set at 520 eV, and the electronic energy was converged to less than $10^{-5}$ eV in all calculations.

**Film fabrication and characterization**. Thin films of BFO, Mn (5%)-doped BFO (Mn-BFO) and Ni (5%)-doped BFO (Ni-BFO) with a thickness of 300 nm were deposited on vicinal SrTiO₃ (STO) (100) single-crystal substrates with a miscut angle of 4° along the [110] direction by pulsed-laser deposition with a KrF excimer laser ($\lambda=248$ nm). We set the following parameters for the film preparation: a substrate temperature of 630 °C, an oxygen pressure of 11 Pa, a laser energy of 1.2 J cm⁻² and a laser repetition rate of 7 Hz. Domain structures were investigated by piezoresponse force microscopy (PFM) and X-ray diffraction measurements. For the PFM observations, in-plane and out-of-plane phase and amplitude images were measured.

**PV measurements**. The PV properties of the SD films along the $y$ and $-x$ directions were measured under illumination with light at wavelengths of 405 nm ($h\nu=3.1$ eV), 515 nm ($h\nu=2.4$ eV) and 639 nm ($h\nu=1.9$ eV). The light propagated along the $z$ axis and was incident on the sample surface. The power of the incident light was set at 2.5 W cm⁻². The incident light was adjusted to have a polarization angle ($\omega$) from the $y$ axis in the $xy$ plane using a polarizing plate and a half-wave plate. Pseudocubic notation is adopted throughout this article to express the crystallographic directions.

The BPV tensor elements ($\beta_{15}$, $\beta_{22}$, $\beta_{31}$ and $\beta_{33}$) were estimated from the $\omega$ dependences of $J_y$ and $J_{-x}$. When linearly polarized light is incident on a ferroelectric crystal, the photocurrent density $\mathbf{J}$ derived from the BPV effect is expressed by the equation

$$\mathbf{J}_i=I_{\text{opt}}\beta_{ijk}\mathbf{e}_j\mathbf{e}_k,\qquad(2)$$

where $\beta_{ijk}$ is a third-rank BPV tensor, $I_{\text{opt}}$ is the incident light intensity, and $\mathbf{e}_j$ and $\mathbf{e}_k$ are the components of the unit vectors along the $j$ and $k$ directions in the crystallographic coordinate system, respectively. For ferroelectrics in $C_{3v}$ symmetry, $\beta_{ijk}$ in matrix notation is expressed as

$$\boldsymbol{\beta}_{ijk}=\begin{pmatrix} 0 & 0 & 0 & 0 & \beta_{15} & -\beta_{22} \\ -\beta_{22} & \beta_{22} & 0 & \beta_{15} & 0 & 0 \\ \beta_{31} & \beta_{31} & \beta_{33} & 0 & 0 & 0 \end{pmatrix}.\qquad(3)$$

We obtain $J_{[110]}$ and $J_{[\bar{1}10]}$ as follows:

$$\begin{aligned}J_{[110]}=&\frac{I_{\text{opt}}}{3\sqrt{3}}\left(\sqrt{2}\beta_{15}+\beta_{22}+2\sqrt{2}\beta_{31}+\sqrt{2}\beta_{33}\right)\\&+\frac{I_{\text{opt}}}{3\sqrt{3}}\left(\sqrt{2}\beta_{15}-2\beta_{22}-\sqrt{2}\beta_{31}+\sqrt{2}\beta_{33}\right)\sin\left[2\left(\omega+\frac{\pi}{4}\right)\right].\end{aligned}\qquad(4)$$

$$J_{[\bar{1}10]}=I_{\text{opt}}\left[\frac{2}{\sqrt{6}}\beta_{15}+\frac{1}{\sqrt{3}}\beta_{22}\right]\sin(2\omega)\qquad(5)$$

When we neglect the deviation of the measurement directions from the crystallographic orientations and the decay of light in the ferroelectric layer by absorption, we can regard $J_{[110]}$ as $J_y$ and $J_{[\bar{1}10]}$ as $J_{-x}$. The details of the accurate descriptions of $J_y$ and $J_{-x}$ taking into account the light absorption and the vicinal angle are described in ref. [32]. For the calculations of the PV tensor elements, we adopted the relation $\beta_{22}=1/10\beta_{33}$ that has been proven for LiNbO₃ in $C_{3v}$ symmetry.

**Data Availability**. The data that support the findings of this study are available from the corresponding author on reasonable request.

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

## Acknowledgements

This research is supported by JSPS through a Grant-in-Aid for JSPS Fellows (14J04693), partly by JSPS KAKENHI Grant Number 26249094. We thank T. Numata (HORIBA Techno Service Co., Ltd) for photoluminescence measurements and K. Yoshida for

assistance with PV measurements. We acknowledge R. Inoue and Y. Kitanaka for fruitful discussion.

## Author contributions

H.M. and Y.N. conceived and designed the experiments. H.M. performed the experiments and the tensor analyses. Y.N. calculated the electronic structures. H.M., Y.N. and M.M. wrote the paper.

## Additional information

**Competing interests:** The authors declare no competing financial interests.

