## [Peer Review File · Nature Communications]

Reviewers' comments:

Reviewer #1 (Remarks to the Author):

The manuscript "Gap-state engineering of visible-light-active ferroelectrics for photovoltaic applications" reports on the observation and tuning of the bulk-photovoltaic effect (BPE) in BiFeO₃ crystals, by the introduction of states in the band gap of the material, in order to overcome the inherent problem of optical transparency of the parent crystal, without sacrificing polarisation.

I found that the manuscript was well presented and clear and that the results demonstrate the effectiveness of the researcher's strategy to boost the bulk photovoltaic effect. However, I have several issues with the paper that mean that it would require significant revision before I could consider accepting it for publication in Nature Communications.

The majority of my contention stems from the fact that the concept of introducing gap states to improve photovoltaic absorption/performance is nothing new, despite the authors claims to the contrary. There is a history of over a decade of research into so-called intermediate band solar cells (IBSC), which operate on precisely the principle that the authors claim to be a 'new-approach'. First and foremost I would suggest that the authors review this field more thoroughly: https://en.wikipedia.org/wiki/Intermediate_band_photovoltaics

My next contention is related to the well known problems in the field of IBSC - poor carrier lifetimes. By definition the gap state that is introduced to enhance absorption is also a killer defect state for Shockley-Read-Hall recombination. This is an issue which has plagued IBSC (doi:10.1038/nphoton.2012.1). For this reason I do not consider that reports of an enhanced BPE in the material is sufficient to warrant getting over-excited. In order to assess if this approach will truly enhance PV performance. I would very much like to see some data on carrier lifetimes in the pristine and gap-state samples, to see if the gain from additional absorption is negated by increased recombination or not.

Reviewer #2 (Remarks to the Author):

This is a nicely written , detailed article on enhancing the PV response of ferroelectric oxides, such as BFO, by carefully engineering the mid-gap states. In this case, the authors have used Mn doping to accomplish this. The paper is very thoughtfully written and I think with some edits it can be accepted.

I found the PFM data not very convincing; indeed it seems not too relevant to the thesis of the paper and so could be deleted. SO also, I thought the paper could benefit with some tightening of the prose and elimination of some peripheral details.

After this, I think the paper can be accepted.

Reviewer #3(Remarks to the Author):

In their manuscript the authors first investigate the electronic structure of BiFeO₃ (BFO) doped with different transition metal (TM) elements on the Fe-site by density functional theory. They find that Cr, Mn and Co create 3d electronic states at different energetic positions in the band gap, half of which are filled with electrons; the corresponding states of Ni lie within the valence band, causing the appearance of unfilled oxygen shallow 2p-states in the band gap. Doping with Cu leads to the appearance of unfilled 3d-states in the gap. It is suggested that the TM creating half-filled states near the center of the gap favour bulk photovoltaic charge transport under sub-bandgap illumination. To test this hypothesis, films of BFO doped with 5% of Mn. These samples show a significant increase of the components of the bulk photovoltaic tensor when illuminated with photon energies of 2.4 eV, i.e. lower than the band gap of 2.7 eV observed by

Response letter

Yuji Noguchi (The University of Tokyo)

To the 1st reviewer

Thank you very much for your kind and useful comments on our paper. I appreciate all of your comments. I would like to respond each of your comments below.

Comment 1

The majority of my contention stems from the fact that the concept of introducing gap states to improve photovoltaic absorption/performance is nothing new, despite the authors claims to the contrary. There is a history of over a decade of research into so-called intermediate band solar cells (IBSC), which operate on precisely the principle that the authors claim to be a ‘new-approach’. First and foremost I would suggest that the authors review this field more thoroughly: https://en.wikipedia.org/wiki/Intermediate_band_photovoltaics

Reply 1

We agree with the point that the gap state plays a similar role as a stepping stone of IBs in IBSCs regarding the enhanced optical absorption of visible light. We also think that the *similarity* and *difference* of the gap-state engineering with the IBs need to be described clearly. Therefore, we have added the short review of IBSCs in the beginning of **Introduction**, as shown below.

The beginning of **Introduction** (page 3)

Innovative strategies have recently been introduced into photovoltaics for efficient solar energy conversion^{1,2,3,4}. To absorb light over a broad spectral range and convert the incident energy into electricity, intermediate band (IB) solar cells are designed, in which the IB provides electronic states that are partially filled with electrons within the bandgap of a host semiconductor. The IB acts as a stepping stone to generate an electron-hole pair under illumination with below bandgap energy, thereby delivering a large photocurrent^{5,6,7}. In spite of extensive studies on IB solar cells, the present devices suffer from a reduction in open-circuit voltage compared with the reference cells without IBs⁸.

We would like to emphasize that there is a big difference between the gap-state engineering and the IBs in their respective impacts on the photovoltaic properties. At present, IBSCs suffer from a reduction in open-circuit voltage at room temperature while showing a larger photocurrent (doi: 10.1038/nphoton.2012.1). A strong advantage of the gap-state engineering in ferroelectric is that not only photocurrents but also photovoltages are enhanced markedly over a broad spectral range (regardless of the above- or below-bandgap light: please see **Fig. 4**). Our

conclusion is that the major origin of the enhanced PV response achieved by the gap-state engineering is *not the increase in absorption coefficient but the large photogeneration strength* (much larger Glass coefficients), which is quite different from the role of the IBs in IBSCs. To clearly show the advantage of the gap-state engineering and its difference with the IBs, we have modified the main text, as listed below.

Abstract

Our first-principles calculations and single-domain thin-film experiments of BiFeO₃ demonstrate that gap states half-filled with electrons can enhance not only photocurrents but also photovoltages over a broad photon-energy range, which is different from intermediate bands in present semiconductor-based solar cells.

The last part of Introduction (page 5)

Our theoretical and experimental investigations demonstrate that the gap states enhance both photocurrents and photovoltages over a broad spectral range.

Discussion (page 15)

These results lead to the conclusion that the major origin of the enhanced PV response achieved by the Mn doping is not the increase in α but the large Glass coefficients.

The last part of Discussion (page 15)

Finally, we address how the half-filled gap states in ferroelectrics are different from IBs in semiconductor solar cells. For bulk IB solar cells, such as V₂In₁₄S₂₄⁵⁵ and ZnTe:O⁷, extended IBs are situated in or near the middle of the band gap. The IBs act as a stepping stone for generating electron-hole pairs by absorbing sub-bandgap photons and thereby photocurrents are enhanced by an increase in α at $h\nu < E_g$ ⁸. The crucial problem to be overcome is a reduction in V_{oc} with respect to the reference cells without IBs at room temperature⁸, where the Shockley–Read–Hall (SRH) recombination⁵⁶ influences their V_{oc} to some extent. In contrast, the gap state in ferroelectrics can not only increase α but also enhance V_{oc} . In fact, the Mn doping leads to larger values both of V_{oc} and J_{sc} (Fig. 4). Although the SRH recombination is suggested to reduce the carrier lifetime, the gap state increases the Glass coefficients by over one order of magnitude at $h\nu = 2.4$ eV ($< E_g$). This experimental fact shows that the average drift velocity of photogenerated electrons and holes is higher in the presence of the gap state, and thereby delivering the robust BPV effect.

Comment 2

My next contention is related to the well-known problems in the field of IBSC - poor carrier lifetimes. By definition the gap state that is introduced to enhance absorption is also a killer defect state for Shockley-Read-Hall recombination. This is an issue which has plagued IBSC (doi:10.1038/nphoton.2012.1). For this reason, I do not consider that reports of an enhanced BPE in the material is sufficient to warrant getting over-excited. In order to assess if this approach will truly enhance PV performance. I would very much like to see some data on

carrier lifetimes in the pristine and gap-state samples, to see if the gain from additional absorption is negated by increased recombination or not.

Reply 2

We performed photoluminescence (PL) measurements and subsequently had planned to conduct fluorescence lifetime observations to get information on carrier lifetimes in our samples with the help of a technical staff of HORIBA, Ltd. We measured the PL spectra by using the laser Raman spectrometer (LabRAM HR Evolution) that enables us to measure photoemissions with an ultrahigh sensitivity. Film samples (on SrTiO₃ substrates) exhibit a strong peak at 3.1 eV attributed to the bandgap emission of SrTiO₃, which completely covers the signal from the ferroelectric films. Therefore, we conducted the PL measurements for the dense ceramics. Since the signal of the emissions was very weak at room temperature, the data were collected at 77 K. The low-temperature PL spectra are shown in **Response Figure 1**. We found emissions at ~ 3 eV and ~ 2.4 eV for BiFeO₃. The peak at ~ 3 eV is ascribed to a *dipole-allowed* charge transfer (CT) transition, and that at ~ 2.4 eV is assigned to a *dipole-forbidden* CT transition (doi:10.1103/PhysRevB.79.235128). The emission at ~ 2.4 eV agrees well with the PRB paper (doi: 10.1103/PhysRevB.89.035133), where the carrier lifetime is estimated to be ~ 100 ns.

The Bi(Fe_{0.95}Mn_{0.05})O₃ ceramic also exhibits these two peaks. Unfortunately, we cannot detect the apparent signal associated with sub-bandgap emissions arising from the Mn-derived gap state. Even if the tail in the range of 1.6–1.9 eV contained information on a sub-bandgap emission, the carrier lifetime measurement requires a signal that is three orders of magnitude larger than that observed. These results force us to give up fluorescence lifetime measurements.

Response Figure 1. Photoluminescence (PL) spectra observed for the ceramic samples of BFO and Mn (5%)-BFO at 77 K by using the laser Raman spectrometer (LabRAM HR Evolution, HORIBA, Ltd.). Emissions excited by a continuous wave HeCd laser [5 mW at 325 nm (3.8 eV)] were collected in a backscattering configuration at different spots on the surface, where a typical spot size is 0.3 mm in diameter. The data averaged over four spots are plotted. We thank Tomoko Numata at HORIBA Techno Service Co., Ltd. for the measurements of the PL spectra.

We would like to emphasize that the conclusion in our paper remains valid and that the major part of the discussion is not affected, even if information on the carrier lifetimes is missing, because the experimental results displayed in **Fig. 4** undoubtedly show that the gap state resulting from the Mn doping in BFO enhances not only photocurrents but also photovoltages in a broad spectral range. We have added the brief discussion regarding the possible influence of the SRH recombination associated with reduced carrier lifetimes in the end of Discussion.

The emission intensity (count/s: cps) measured for the Mn-BFO ceramic is much smaller than that of the BFO ceramic. These results suggest that the lifetimes of photogenerated electrons in the CB and holes in the VB are shortened by nonradiative recombinations in the presence of the Mn-derived gap state. Even though the carrier lifetimes are shortened, the photogeneration strength (the Glass coefficients) is markedly enhanced for the Mn-BFO film, as indicated in **Fig. 4**.

To the 2nd reviewer

Thank you very much for your kind and useful comments on our paper. I appreciate your comments. I would like to respond each of your comments below.

Comment 1

I found the PFM data not very convincing; indeed it seems not too relevant to the thesis of the paper and so could be deleted. SO also, I thought the paper could benefit with some tightening of the prose and elimination of some peripheral details. After this, I think the paper can be accepted.

Reply 1

We agree with the reviewer's suggestion. The data of the PFM and XRD associated with their related sentences describing the single-domain state of the films have been moved to **Supplementary Figure 2**. In addition, the peripheral details with ~ 200 words have been moved to the captions of **Supplementary Figures** and those with ~ 100 words were deleted from the main text.

To the 3rd reviewer

Thank you very much for your kind and useful comments on our paper. I appreciate all of your comments. I would like to respond each of your comments below.

Comment 1

The main claim is that it is not necessary to illuminate with above-bandgap energies in the doped BFO, as electron-hole pairs are generated by first exciting the hole by lifting an electron from the valence band into an empty gap state with sub-bandgap energies, and then exciting it further into the conduction band with a second photon of the same sub-bandgap energy. This claim is novel, but there are problems with the evidence that is presented:

Why is it necessary to generate an electron-hole pair in the first place? The prototypical system displaying the bulk photovoltaic effect is LiNbO₃. The band gap of this system is above 3 eV, and at 532 nm it shows neither notable absorption nor photocurrent. Doping with Fe creates filled donor levels in the band gap; now, illumination with 532 nm raises electrons from these donor levels into the conduction band where it then travels, creating a significant photocurrent and leaving behind an unfilled trap that will be re-filled by capturing another photoexcited electron. The center is then available for the next passing photon to re-excite the electron. There are no holes involved in the process. In the present case, it is interesting that Cr, Mn and Co can potentially act as both donors and acceptors, while Ni and Cu are exclusively acceptors. However, is it really necessary to assume that both electrons and holes play a significant role in the charge transport? How can it be excluded that, e.g., excitation of only either electrons or holes from the centers (then acting exclusively as donors or acceptors, respectively) is sufficient to create the observed photovoltaic effect, without the involvement of holes? In this case, there would be no two-photon process required to explain the results, and it would just be the case of donor or acceptor doping to increase the photovoltaic effect, which is well-described in literature.

Reply 1

To answer the question regarding majority photocarriers, we have conducted the following additional experiments: one is the fabrication and the PV investigation of the Ni-doped BFO film (new sample) and the other is the PV measurements of the Mn-BFO film at $h\nu = 1.5$ eV. Moreover, to get further insight into the optical transition process via the Mn-derived gap state, DFT calculations were performed to investigate the energy levels of the gap states of Mn⁴⁺ and Mn²⁺, because the carrier-generation process mediated through the gap state is accompanied by the change in the valence state of Mn.

Supplementary Figure 9 indicates the experimental results obtained for the Ni-BFO film. The marked increase in dark conductivity (σ_{dark}) by an O_3 annealing provides a clear indication of the Ni-BFO film having an empty state in the vicinity of the VBM, the details of which are described in **Supplementary Table 3** and **Supplementary Note 3**.

As shown in **Fig. 1**, Ni-BFO is a representative example that illumination with $h\nu = 1.9$ eV is expected to induce the BPV effect arising from photogenerated holes as a majority carrier, because these low-energy photons can pump electron only from the VB to the empty gap state (cannot pump electron from the gap state to the CBM owing to a large energy difference of ~ 2.7 eV). The PV measurements show that the Ni-BFO film displays a high photocurrent at $h\nu = 1.9$ eV. These results lead to the conclusion that the PV response observed for the Ni-BFO film stems from the BPV effect resulting from photogenerated holes, i.e., the majority photocarrier is hole. Therefore, as pointed out by the reviewer, we recognize that the BPV effect can be induced by the generation of not only hole-electron pairs but also one type of carrier (either photogenerated electron or hole).

As for Mn-BFO, taking account of the energy level of the Mn^{3+} state that is located at 0.9 eV from the VBM and is situated at 1.9 eV with respect to the CBM, we expected that light in the energy range of 1.0 eV–1.8 eV is capable of delivering the BPV effect arising from photogenerated holes. To test this hypothesis, we have performed the PV measurements at $h\nu = 1.5$ eV (please see **Supplementary Figure 12**). The results indicate that a short-circuit photocurrent at $h\nu = 1.5$ eV was below the detection limit. In contrast, the photoconductivity (σ_{photo}) at $h\nu = 1.5$ eV is indeed higher than the σ_{dark} (**Supplementary Table 3**). The experimental fact of the higher σ_{photo} shows that the photon absorption leaving behind holes takes place but the photogenerated holes do not act as an effective carrier for the BPV effect. The possible reason why the photogenerated holes do not lead to the BPV effect is described in **Supplementary Note 5**.

As shown in **Fig. 4**, the Mn-BFO film exhibits an apparent PV effect at $h\nu = 1.9$ eV. Under illumination with $h\nu = 1.9$ eV, the optical transition associated with the Mn^{3+} – Mn^{2+} change injects holes in the VB. Because this optical process is essentially the same at $h\nu = 1.5$ eV (at which the photogenerated holes alone do not deliver a detectable BPV effect), it is unlikely that the BPV effect at $h\nu = 1.9$ eV stems from photogenerated holes. The photons at $h\nu = 1.9$ eV can pump electron from the Mn^{3+} state to the CB (the Mn^{3+} – Mn^{4+} change) and further pump electron from the VB to the Mn^{4+} state (going back to the ground state with Mn^{3+}). This successive optical process can generate hole-electron pairs. These results permit us to consider that the BPV effect at $h\nu = 1.9$ eV observed for the Mn-BFO film originates from the generation of hole-electron pairs.

As the reviewer points out, the BPV effect arising from the excitation of either electrons

or holes from the gap state (one-photon process) has been reported in some ferroelectric materials such as Fe-doped LiNbO₃ [Phys Status Solidi **113**, 157-164 (1982), Phys. Status Solidi **212**, 2968-2974 (2015), Adv. Condens. Matter Phys. **2016** (2016)] and Mn-doped BaTiO₃ [Jpn. J. Appl. Phys **52**, 09KF03 (2013)]. Moreover, the Ni-BFO film indeed exhibits the BPV effect at 1.9 eV derived from photogenerated hole as a majority carrier. Considering the above described results together with the previous reports, we have revised our manuscript largely, as shown below. In addition, the story from **Introduction** to **Discussion** is modified to describe that we choose Mn and Ni as a dopant for differentiating the influences of the half-filled gap state and the empty state on the BPV effect. Furthermore, an index representing photogenerated power, $|J_y \times V_y|/4$, has been added to compare the BPV effect observed for the films of BFO, Mn-BFO, and Ni-BFO in **Supplementary Figure 9 (c)**.

Results

Electronic structure calculations (page 7–8)

For gap states filled with electrons, pumping electrons from the gap states to the CB injects carriers leaving behind empty traps, and the continuous illumination yields a PV current arising from a sequential trapping and detrapping of photogenerated electrons^{21,22,49}. When gap states are empty, pumping electrons from the VB to the gap states creates holes, and this illumination also brings a PV current via a trapping and detrapping of photogenerated holes²⁶.

Provided that the gap states are partially filled with electrons, we expect that the absorption of two sub-bandgap photons generates electron-hole pairs by pumping electrons from the VB to the CB being mediated through the gap states, as is seen for IB solar cells^{3,8}. This two-photon absorption has a high probability when its state is half-filled⁵⁰. Moreover, its energy level is preferred in or near the middle of the bandgap. Compared with the filled or empty gap states, where the majority photocarrier is either electron or hole, respectively, we anticipate that the half-filled mid-gap state acts as a scaffold for a robust PV response under illumination with below-bandgap light.

Discussion (page 12–13)

The Mn-BFO film exhibits an apparent PV response at $h\nu = 1.9$ eV but no detectable photocurrent at $h\nu = 1.5$ eV. These results combined with the optical transition processes [**Supplementary Note 5**] suggest that the majority photocarrier at $h\nu = 1.9$ eV is electron-hole pair. In contrast, we found that photogenerated holes can induce the BPV effect in the Ni-BFO film at $h\nu = 1.9$ eV [**Supplementary Fig. 9**]. The Ni doping provides the empty state in the vicinity of the VBM. This shallow state gives rise to the high conductivities under dark and illumination [**Supplementary Table 3**], leading to the high currents with sacrificing the voltages. The much higher photogenerated powder of the Mn-BFO film compared with the Ni-BFO film show that the half-filled gap state is effective for the BPV effect.

Comment 2

An indication that the creation of an electron-hole pair is actually not required for the observed effect is the ratio of the tensor elements shown in Fig. 6. In undoped BFO, creation of such a pair is indeed the only way to achieve charge transport, as there is no other source of electrons

except the valence band. In this case, β_{33} is negative at both 2.4 eV and 3.1 eV, while β_{15} and β_{31} are positive. The direction in which the electrons and holes move is independent of the excitation energy, which makes sense. This changes in the doped system. Here, the sign of beta changes compared to the undoped system for β_{15} when exciting at 3.1 eV, and for β_{31} when exciting at 2.4 eV. Apparently, either the ratio of the number of the two carrier types changes, or the direction that electrons and holes travel now depends on the excitation wavelength. The first explanation appears more plausible, which means there is no even creation of electrons and holes any more. One carrier type will dominate the other, and the dopant is mainly acting as either donor or acceptor. There is no two-photon process involved in this. Even if the second explanation would be true, it means that the dopant is not merely acting as a stepping stone in the creation of electron-hole pairs: electrons/holes excited from the dopant behave differently than those excited from the valence/conduction band transition.

Reply 2

The bulk PV (BPV) effect in ferroelectric oxides has been understood by the shift current theory [Phys. Rev. B **61**, 5337 (2000), Phys. Rev. Lett. **109**, 236601 (2012), Phys. Rev. Lett. **109**, 116601 (2012)]. The shift current in polar materials with spatial inversion symmetry breaking arises from the second-order nonlinear optical response [Phys. Rev. B **61**, 5337 (2000), Phys. Rev. B **94**, 035117 (2016)]. The experimental data observed for BaTiO₃ single crystals have demonstrated that not only the strength but also the direction of photocurrents are strongly dependent on photon energy of incident light [Ferroelectrics **13**, 305 (1976)]. This behavior of photocurrents is well explained by the shift current theory [Phys. Rev. Lett. **109**, 116601 (2012)], where the tensor elements of the BPV effect exhibit complicated tendencies on photon energy and the sign of them also changes. The similar behavior has been predicted also for BiFeO₃ under above-bandgap light. The sign of the shift current is determined by the shift vector that is governed by the details of the electronic structure including the Berry connection [Phys. Rev. Lett. **109**, 236601 (2012)].

It is reasonable to consider that the shift current associated with the Mn-derived gap state displays a distinct feature, which is different from that caused by the VB-CB transition. The shift current theory taking account of gap states is required for explaining the details of the BPV tensor elements listed in **Supplementary Tables**. We think that it is beyond the target of our manuscript.

Comment 3

One way to resolve this question could be to perform Hall measurements of the photocurrent, which should reveal if the majority carrier type changes with doping. Another option would be

to measure the photovoltaic effect in the doped system with wavelengths that allow excitation of only one carrier type. For example, in the Mn-doped system illumination of 1 eV or slightly above should be sufficient to excite holes by lifting electrons from the valence band into the dopant level, but not further excite the electrons to the conduction band. If this already results in a significant increase of the photovoltaic effect compared to the undoped sample at the same wavelength, then the two-photon mechanism is not required. Ideally, the same measurement should also be done by exciting only electrons from the Mn-center, but this is not possible, as illumination with 1.9 eV or above will also excite holes.

Reply 3

We think that Hall measurements provide direct evidence of photogenerated holes or electrons delivering the BPV effect. Unfortunately, we cannot access the special equipment that enables us to observe Hall effects under illumination (as far as we are aware, only the following paper [JJAP, 15(11), 2263 (1976)] provides direct evidence of photogenerated electron as a majority carrier for reduced LiNbO₃ crystals under Xe-ark lamp, which was revealed by photo Hall effect measurements).

As answered in **Reply 1**, we have prepared a new sample of the Ni-BFO film and performed the PV measurements. The experimental results demonstrate that the BPV effect at $h\nu = 1.9$ eV stems from photogenerated holes that act as a majority carrier (please see **Supplementary Figure 9**, **Supplementary Table 3**, and **Supplementary Note 3**). As a result, we came to the conclusion that either photogenerated electron or hole acts as a majority carrier leading to the BPV effect, which is a minor part in our manuscript.

As to the Mn-BFO film (as answered in **Reply 2**), we cannot detect a short-circuit photocurrent at $h\nu = 1.5$ eV, although this photon energy is considered sufficient to pump electron from the VB to the gap state. We answer this point in **Reply 1** in detail.

Comment 4

In addition, it is difficult to judge the validity of the numerical calculations based on equation (1), as they depend on the measured values of the absorption. Looking at the supplementary material, the absorption coefficient is still in the order of several 10^4 /cm in the range between 2.1 eV and 2.4 eV even in the undoped system, i.e. far below the band gap. This seems to be far too high for a pure insulating oxide. There are no details given as to how this value was obtained; can it be assured that e.g. reflection correction was done correctly, and that the absorption of the SrTiO₃ substrate was taken into account? More details on the experimental procedure should be given in the supplementary material.

Reply 4

The question regarding the optical absorption in the range between 2.1 eV and 2.4 eV in BFO ($E_g = 2.8$ eV) is quite natural, but this optical absorption has been reported in refs [doi:10.1103/PhysRevB.79.235128, *Phys. Rev. B* **89**, 35133 (2014), *Appl. Phys. Lett.* **96**, 192901 (2010)] and the absolute value of absorption coefficient agrees well with these reports. In addition, the optical transition centered at 2.4 eV is confirmed by the photoluminescence spectra observed for the dense ceramic of BiFeO₃, as presented in **Response Figure. 1**. The absorption at ~ 2.4 eV is assigned to a *dipole-forbidden p-d* charge transfer (CT) transition (doi:10.1103/PhysRevB.79.235128), which is described in the main text. To describe the optical absorption under illumination below E_g , the following sentences have been added in the caption of **Supplementary Figure 3**. As to the method of evaluating absorption coefficient, the details are described in **Supplementary Note 4**.

The caption in **Supplementary Figure 3**

The BFO film has a strong absorption at ~ 3 eV derived from a *dipole-allowed p-d* charge transfer (CT) transition and an absorption at ~ 2.5 eV, which is assigned to a *dipole-forbidden p-d* CT transition⁵. An apparent absorption at ~ 2 eV was also observed, which is tentatively attributed to the unoccupied gap (b_1) state arising from Fe⁴⁺ adjacent to oxygen vacancies as a result of the formation of Bi vacancies⁶. The apparent absorption at ~ 2.5 eV with a tail has been also reported in refs. 7–9.

visible light spectroscopy.

The manuscript is interesting to anyone working with either polar oxides or photovoltaic systems such as perovskites; particularly the latter topic is of high interest to a broader community. The strong point of the manuscript is its combination of DFT studies and experimental verification. The presentation style is clear, and the presentation in the context of other literature is very good. The main claim is that it is not necessary to illuminate with above-bandgap energies in the doped BFO, as electron-hole pairs are generated by first exciting the hole by lifting an electron from the valence band into an empty gap state with sub-bandgap energies, and then exciting it further into the conduction band with a second photon of the same sub-bandgap energy. This claim is novel, but there are problems with the evidence that is presented:

Why is it necessary to generate an electron-hole pair in the first place? The prototypical system displaying the bulk photovoltaic effect is LiNbO_3 . The band gap of this system is above 3 eV, and at 532 nm it shows neither notable absorption nor photocurrent. Doping with Fe creates filled donor levels in the band gap; now, illumination with 532 nm raises electrons from these donor levels into the conduction band where it then travels, creating a significant photocurrent and leaving behind an unfilled trap that will be re-filled by capturing another photoexcited electron. The center is then available for the next passing photon to re-excite the electron. There are no holes involved in the process. In the present case, it is interesting that Cr, Mn and Co can potentially act as both donors and acceptors, while Ni and Cu are exclusively acceptors. However, is it really necessary to assume that both electrons and holes play a significant role in the charge transport? How can it be excluded that, e.g., excitation of only either electrons or holes from the centers (then acting exclusively as donors or acceptors, respectively) is sufficient to create the observed photovoltaic effect, without the involvement of holes? In this case, there would be no two-photon process required to explain the results, and it would just be the case of donor or acceptor doping to increase the photovoltaic effect, which is well-described in literature.

An indication that the creation of an electron-hole pair is actually not required for the observed effect is the ratio of the tensor elements shown in Fig. 6. In undoped BFO, creation of such a pair is indeed the only way to achieve charge transport, as there is no other source of electrons except the valence band. In this case, β_{33} is negative at both 2.4 eV and 3.1 eV, while β_{15} and β_{31} are positive. The direction in which the electrons and holes move is independent of the excitation energy, which makes sense. This changes in the doped system. Here, the sign of beta changes compared to the undoped system for β_{15} when exciting at 3.1 eV, and for β_{31} when exciting at 2.4 eV. Apparently, either the ratio of the number of the two carrier types changes, or the direction that electrons and holes travel now depends on the excitation wavelength. The first explanation appears more plausible, which means there is no even creation of electrons and holes any more. One carrier type will dominate the other, and the dopant is mainly acting as either donor or acceptor. There is no two-photon process involved in this. Even if the second explanation would be true, it means that the dopant is not merely acting as a stepping stone in the creation of electron-hole pairs: electrons/holes excited from the dopant behave differently than those excited from the valence/conduction band transition.

One way to resolve this question could be to perform Hall measurements of the photocurrent, which should reveal if the majority carrier type changes with doping. Another option would be to measure the photovoltaic effect in the doped system with wavelengths that allow excitation of only one carrier type. For example, in the Mn-doped system illumination of 1 eV or slightly above should be sufficient to excite holes by lifting electrons from the valence band into the dopant level, but not further excite the electrons to the conduction band. If this already results in a significant increase of the photovoltaic effect compared to the undoped sample at the same wavelength, then the two-photon mechanism is not required. Ideally, the same measurement should also be done by exciting only electrons from the Mn-center, but this is not possible, as illumination with 1.9 eV

or above will also excite holes.

In addition, it is difficult to judge the validity of the numerical calculations based on equation (1), as they depend on the measured values of the absorption. Looking at the supplementary material, the absorption coefficient is still in the order of several 10^4 /cm in the range between 2.1 eV and 2.4 eV even in the undoped system, i.e. far below the band gap. This seems to be far too high for a pure insulating oxide. There are no details given as to how this value was obtained; can it be assured that e.g. reflection correction was done correctly, and that the absorption of the SrTiO₃ substrate was taken into account? More details on the experimental procedure should be given in the supplementary material.

In summary, the possibility to introduce centers in the system that can potentially act as either donors or acceptors is highly interesting, but the evidence that they actually do fulfil both roles is insufficient. Therefore, I cannot recommend publication of the manuscript in Nature Communications in its present form. If the authors can present convincing arguments or new data that the dual donor/acceptor role of the dopants is indeed relevant for the observed increase of the bulk photovoltaic effect, this assessment will change.

REVIEWERS' COMMENTS:

Reviewer #1 (Remarks to the Author):

I would like to commend the authors on their full response to all referee's comments. They have addressed many of my initial concerns, primarily expanding on the similarities and differences between their approach and the IBSC approach.

I also thank them for their full report on the difficulties of obtaining PL measurements of lifetimes. I consider that they have tried everything possible to satisfy this request. Additionally I feel that the enhanced photovoltage suggests that recombination should not be much worse in their band-state engineered materials.

I find this to be an interesting, provocative and clear piece of science and I am happy to recommend for publication.

Reviewer #3 (Remarks to the Author):

The authors present a number of good arguments that increase the probability that their interpretation of the data based on the excitation of both electrons and holes is correct. In particular, my previous comments 2 and 4 have been dealt with very appropriately. The main open question is with comment 1: it has to be noted that there still is no compelling evidence to show that both types of carriers are involved in the charge transport; however, the approach now appears much more reasonable. I cannot say in how far this approach is novel for semiconductor-based photovoltaics, but in the field of bulk photovoltaic ferroelectrics, the use of a single dopant as both donor and acceptor is an interesting concept. Therefore, I recommend that the manuscript is published in Nature Communications.

Response letter

To the Reviewer #1

Thank you very much for your positive comments on our paper. Our replies are described below your comments.

Reviewer #1 (Remarks to the Author):

#1.1. Comment

I would like to commend the authors on their full response to all referee's comments. They have addressed many of my initial concerns, primarily expanding on the similarities and differences between their approach and the IBSC approach.

#1.1. Reply

Thanks to your comment regarding the IBSC, our revised manuscript has become more mature and will have attract much attention in the broad fields including photovoltaics and polar materials.

#1.2. Comment

I also thank them for their full report on the difficulties of obtaining PL measurements of lifetimes. I consider that they have tried everything possible to satisfy this request. Additionally, I feel that the enhanced photovoltage suggests that recombination should not be much worse in their band-state engineered materials.

#1.2. Reply

As the reviewer pointed out last time, our PL measurements suggest that the carrier lifetimes under illumination are shortened by the introduction of gaps states. I appreciate that the reviewer fairly judges the following point: our gap-state engineering, in reality, enhances the bulk photovoltaic effect even though the carrier recombination plays a role. In our next research target, we will investigate the carrier lifetimes in pristine and gap-state-engineered samples.

#1.3. Comment

I find this to be an interesting, provocative and clear piece of science and I am happy to recommend for publication.

#1.3. Reply

We are glad the reviewer recommends our paper for publication.

To the Reviewer #3

Thank you very much for your positive comments on our paper. Our replies are described below your comments.

Reviewer #3 (Remarks to the Author):

#1.1. Comment

The authors present a number of good arguments that increase the probability that their interpretation of the data based on the excitation of both electrons and holes is correct. In particular, my previous comments 2 and 4 have been dealt with very appropriately. The main open question is with comment 1: it has to be noted that there still is no compelling evidence to show that both types of carriers are involved in the charge transport; however, the approach now appears much more reasonable. I cannot say in how far this approach is novel for semiconductor-based photovoltaics, but in the field of bulk photovoltaic ferroelectrics, the use of a single dopant as both donor and acceptor is an interesting concept.

#1.1. Reply

As the reviewer points out, we do not provide direct, solid evidence that the photovoltaic effect stems from both photogenerated electrons and holes under illumination at present. On the one hand, we appreciate that the reviewer thinks our approach being much more reasonable. As described in the previous **Response Letter**, the concept that the introduction of electron-half-filled states near the middle of the bandgap has already been discussed in semiconductor-based photovoltaic devices as IBSCs, which suffers from the reduction in photovoltage. Our gap-state engineering has a strong advantage over the IBSCs: the gap states do not solely act as a stepping stone as for the intermediate bands in the IBSCs and do as a robust scaffold for enhancing not only photocurrent but also photovoltage. We are glad that the reviewer thinks our gap-state engineering is an interesting concept in the field of bulk photovoltaic ferroelectrics.

#1.2. Comment

Therefore, I recommend that the manuscript is published in Nature Communications.

#1.2. Reply

We are glad the reviewer recommends our paper for publication.